# Exploring the Workplace Bullying of Indonesian Caregivers and Its Influencing Factors in Taiwan

**DOI:** 10.3390/ijerph19084909

**Published:** 2022-04-18

**Authors:** Yun-Ping Lu, Bih-O Lee, Chih-Kuang Liu, Ke-Hsin Chueh

**Affiliations:** 1Department of Nursing, Yeezen General Hospital, Yangmei District, Taoyuan City 326, Taiwan; n9177@yeezen.com.tw; 2College of Nursing, Kaohsiung Medical University, Kaohsiung 80708, Taiwan; biholee@kmu.edu.tw; 3College of Management, Fu Jen Catholic University, New Taipei City 24352, Taiwan; 059435@mail.fju.edu.tw; 4Department of Nursing, College of Medicine, Fu Jen Catholic University, New Taipei City 24205, Taiwan

**Keywords:** Indonesian caregivers, workplace bullying, sleep quality, mental health

## Abstract

Background: Bullying can pose a risk to the health and safety of humans, including the risk of damage to the emotional, psychosocial, mental, or physical health of employees in the workplace. In this study, we aimed to understand the personal characteristics, mental health, sleep quality, and workplace bullying status of Indonesian caregivers and explore the influencing factors of workplace bullying among them. Methods: This cross-sectional study was based on a structured questionnaire in Indonesian, which was designed to collect the data of essential personal characteristics, workplace bullying, sleep quality, and mental health using the Indonesian versions of the Negative Acts Questionnaire–Revised (NAQ-R), Pittsburgh Sleep Quality Index (PSQI), and the Brief Symptoms Rating Scale (BSRS-5). Results: A total of 60.9% of Indonesian caregivers never experienced workplace bullying in Taiwan. A multiple regression analysis revealed that being a household caregiver (*β* = 0.14, *p* = 0.021), sleep quality (*β* = 0.18, *p* = 0.031), and mental health (*β* = 0.44, *p* < 0.001) were significantly correlated with the overall workplace bullying scores of the respondents and revealed that these three variables explained 45% of the variance. Conclusions: Taiwan Indonesian caregivers have a similar workplace bullying rate to Indonesian employees in the workplace. This study indicated the relationships among the workplace bullying of foreign caregivers and demonstrated that being a household caregiver, sleep quality, and mental health were closely related.

## 1. Introduction

Various Asian countries have opened up opportunities for foreign workers and caregivers to work in their countries in recent years. For example, starting in April 2019, the Immigration Control and Refugee Recognition Act of Japan has allowed foreign workers and caregivers from Vietnam, the Philippines, Indonesia, Thailand, Cambodia, China, Myanmar, Nepal, and Mongolia to work in Japan [1]. As of the end of October 2020, 1,724,328 foreign migrant workers were working in Japan and this figure represented an increase of 4% compared with that of the previous year [2]. In Malaysia, foreign migrant workers account for 15% of the entire workforce and most of them come from Indonesia (39%), Nepal (24%), Bangladesh (13%), and Myanmar (7%) [3]. Over 300,000 housekeepers are currently working in Hong Kong and more than 290,000 are foreign migrant workers, of whom 49% are from Indonesia and 48% are from the Philippines [4]. Due to better pay, living conditions, and safety, Singapore is a favored destination for foreign migrant workers [5]. The working population of Singapore accounts for 67% of its population of 5.5 million and foreign migrant workers account for 37%—more than one-third—of this workforce [5,6]. At the end of March 2020, Taiwan had 719,000 foreign migrant workers, increasing 1.9% year over year [7].

As foreign workers enter the workplace, they form an important part of the labor force of a country. Due to different languages and cultures, whether foreign migrant workers are adequately protected and whether the working environment is friendly are issues that should be taken seriously by the host countries. When foreign migrant workers are non-citizens, the lack of legal protection in the country where they are working as well as the different culture and customs of the country make foreign migrant workers particularly vulnerable to workplace bullying [8]. Among Pakistani migrant workers working in the United Arab Emirates, 92.5% of respondents mentioned that they had experienced bullying in the workplace [9]. Taiwan allowed foreign caregivers in 1992 and these foreign workers have gradually become an important element of long-term care in Taiwan; their work tasks are not limited to the household care of the elderly and disabled and include everyday services for the family members of their employers [10]. The mental health and workplace situation of foreign caregivers in Taiwan is worthy of in-depth studies and the results can provide a reference for long-term care managers and relevant administrative agencies.

Workplace bullying has been defined as repeated, injurious abuse, including threats, insults, intimidation, or interference with work as well as verbal attacks where the victims and perpetrators may be one or multiple individuals [11]. One of the chief causes of workplace bullying is when employees are forced to accept unreasonable deadlines for their work or are improperly forced to change their work tasks. In addition, being isolated or excluded at work is also a common form of bullying; this isolation or exclusion may imply that personnel from other organizations must accept excessive levels of unnecessary and potentially harmful behavior as a consequence of a long-term unequal balance of power [12]. Most instances of workplace bullying are caused by supervisors and colleagues and the most common form is verbal bullying [13]. Workplace bullying is irrational behavior that is harmful to individuals and groups and is also the cause of health and safety risks; it may consist of verbal attacks, exclusion, harassment, intimidation, the prevention of employees from completing their work, intentionally changing the work content, and withholding information [14].

Workplace bullying behavior can be classified as work task attacks, social intimidation, and personal attacks. Among these types, work task attacks consist of the direct use of work-related attacks (such as withholding information, repeated reminders of faults, intensive inspections, and intentional changes in work tasks) to make it difficult for the victim to complete his or her work, thereby harming the victim [15]. Social and personal attacks involve attacks on an individual for the purpose of causing humiliation or isolation [16].

In the healthcare environment of Japan, three phenomena commonly associated with workplace bullying are the withholding of information, the assignment of excessively large workloads, and insulting or offensive language concerning the individual or the private life of the individual [17]. Younger individuals are more likely to experience bullying [18,19]. Women are more likely to be the victims of bullying [20] but the level of education is uncorrelated with the likelihood of being bullied [19]. Unmarried victims of bullying tend to have stronger emotional reactions to bullying than married victims [21]. People who work night shifts more frequently experience bullying than those who work daytime shifts [22].

Workplace bullying is considered to be a mental health issue. The health of victims is affected by bullying and their perceived health is significantly correlated with whether subjects have encountered bullying [14]. Victims of workplace violence tend to have poor physical and mental health, sleep problems, perceived poor health, and emotional distress, including depression, anxiety, and anger [23]. The longer individuals are subject to bullying, the greater their risk of depression [24]. The more often employees experience perceived workplace bullying, the worse their state of mental health [21]. Depression is the most common symptom of workplace bullying [20].

The percentage of Taiwan households with foreign workers who take over as primary caregivers increased from 12.5% in 1993 to 36.2% in 2019 [7]. In addition, the number of foreign caregivers employed in Taiwan increased steadily from 165,898 in 2008 to 260,188 in November 2019 [7]. Foreign caregivers in Taiwan include household caregivers and institutional caregivers [7]. Foreign caregivers may have to work two or three shifts in a day and their working hours often vary, which commonly leaves them with little free time to participate in social activities; far from home and with no time to spend with their families, foreign caregivers often feel a growing sense of alienation [25]. Furthermore, due to language barriers, foreign caregivers commonly experience poor communication with local residents, family members, and colleagues [26]. Considering these circumstances, whether foreign caregivers working in a social environment with a different language and customs are exposed to workplace bullying is especially worthy of study.

Based on the above descriptions, workplace bullying, sleep quality, and mental health are three important issues among Indonesian caregivers in Taiwan. The relationships among the three variables remain inconsistent [7,20,23]. Therefore, in this study, we aimed to explore workplace bullying and its significant influencing factors among Indonesian caregivers. The results can serve as a reference for long-term care managers and relevant administrative agencies.

## 2. Materials and Methods

### 2.1. Design and Settings

Employing a cross-sectional approach to convenience the sampling, we recruited the subjects of Indonesian caregivers in northern Taiwan for this study. We arranged a quiet room for the employee at hospitals, institutions, and other workplaces or prepared a quiet place for the household caregivers at a community site or at home to explain the purpose of the study, the process, and to obtain consent before the cases were admitted. The questionnaire survey was reviewed and approved by the Taipei City Hospital Institutional Review Board (TCHIRB-10908006-E). Anonymous data collection and coding were performed with the informed consent of the subjects from 5 October 2020 to 26 November 2020.

### 2.2. Subjects

The Indonesian caregivers in Taiwan were recruited in person from three healthcare organizations taken from one hospital, two institutions, and two community care stations. A total of 181 questionnaires were distributed and 179 valid questionnaires were obtained (98.8%). G*Power statistical software (version 3.1.9.7) from California, USA was used; the statistical power was set at a moderate level of 0.15 based on the published literature [27]. The level of significance was set as 0.05 and the power was set at 0.8; the required sample size was at least 123 people.

### 2.3. Research Instrument

The research instrument used in this study was a structured questionnaire in Indonesian; the questionnaire was designed to collect data on personal characteristics, workplace bullying, sleep quality, and mental health. The following personal characteristics were collected: age; marital status; level of education; length of time in Taiwan; types of care services; and whether the subject worked at night.

Based on the Negative Acts Questionnaire–Revised (NAQ-R) developed by Einarsen et al. [28], workplace bullying was classified into three sub-scales: work-related bullying, personal-related bullying, and intimidation toward a person. The questionnaire contained 22 items and the respondents were asked to select the frequency of negative acts encountered during the most recent six months. Responses were scored on a 5-point scale based on 5 choices: never (1 point); sometimes (2 points); roughly once each month (3 points); roughly once each week (4 points); and roughly once each day (5 points). Cronbach’s α (reliability) for the original scale was 0.89 [28] and 0.90 for the Indonesian version [15]. According to the suggestion of the clinical experts, the original item “Practical jokes carried out by people you do not get on with” was deemed unsuitable and was deleted. Finally, the revised sub-scales of the Indonesian NAQ-R were four items of intimidation toward a person, seven items of work-related bullying, and ten items of personal-related bullying. Expert validity testing of the questionnaire yielded a content validity index (CVI) of 100% with a Cronbach’s α of 0.91.

Sleep quality was assessed using the Pittsburgh Sleep Quality Index (PSQI) developed by Buysse et al. [27]. This scale contained 19 items and the respondents were asked to select responses indicating their sleep quality during the most recent month. The items covered the following seven components: subjective sleep quality; sleep latency; sleep duration; habitual sleep efficiency; sleep disturbances; use of sleeping medication; and daytime dysfunction. The overall possible score ranged from 0 to 21 points with a higher score indicating worse sleep quality. The cut-off point for poor sleep quality was ≥ 6 points [29]. With regard to reliability, the original scale had a Cronbach’s α value of 0.83 [27]; the scale used in this study had a Cronbach’s α value of 0.79.

Mental health was assessed using the Brief Symptoms Rating Scale (BSRS-5) developed by Lee et al. [30]. The respondents were asked to select responses indicating their state of mental health during the most recent week. Five items were addressed: anxiety; depression; hostility; sleep disturbance; and sense of inferiority. The overall possible score ranged from 0 to 20 points with a higher score indicating worse mental health. The cut-off point for poor mental health was ≥ 6 points. The original scale had a Cronbach’s α value of 0.77–0.90 and a test–retest reliability of 0.82 [30]. The scale used in this study had a Cronbach’s α value of 0.85.

## 3. Statistical Analysis

After data coding and a file creation, SPSS/Windows 22 software (IBM Corp., Armonk, NY, USA) was used to perform the descriptive and inferential statistical analyses (independent sample *t*-test, one-way analysis of variance (ANOVA), Pearson product–moment correlation, and multiple regression analysis). These analyses were used to explore workplace bullying on the related factors between the variables. The descriptive distribution and inferential analysis of personal characteristics, sleep quality, and mental health with workplace bullying among Indonesian caregivers can be seen in Table 1. The descriptive data of the sub-scales and their items for workplace bullying are provided in Table 2. The multiple regression analysis included independence variables such as age, marriage, education level of junior high school or above, length of time in Taiwan, households, night-time work, night-time on call, sleep quality, and mental health as well as the dependent variable of workplace bullying, as seen in Table 3.

## 4. Results

### 4.1. Distribution and Analysis of Personal Characteristics, Sleep Quality, and Mental Health with Workplace Bullying among Indonesian Caregivers

The subjects who participated in this study were all female Indonesian caregivers with ages ranging from 22 to 51 years (average 35.0 (± 6.8) years). The majority were married (57.5%) and most subjects had an education level of junior high school or above (55.9%). The average length of time in Taiwan was 55.0 (± 32.0) months. Most subjects worked as household caregivers (76.5%) and did not take night-time work (50.8%). The average sleep quality score was 5.2 (± 3.9) points and 34.6% of the subjects had poor sleep quality (PSQI ≥ 6 points). The average mental health score was 2.5 (± 2.9) points and 8.9% of the respondents had poor mental health (BSRS-5 ≥ 6 points) (Table 1).

The age (*r* = 0.16, *p* = 0.031), sleep quality score (*r* = 0.59, *p* < 0.001), and mental health score (*r* = 0.66, *p* < 0.001) of the Indonesian caregivers were significantly positively correlated with workplace bullying. Furthermore, the workplace bullying scores for household caregivers were significantly higher than those for institutional caregivers (*t* = −5.63, *p* < 0.001). A significant difference existed between the overall workplace bullying scores of the respondents and working at night (*F* = 18.67, *p* < 0.001). When the Scheffé test was used to perform a post hoc comparison, the results indicated that the overall workplace bullying scores for the subjects who worked at night were significantly higher than those who did not work at night. The level of education (*F* = 2.81, *p* = 0.063), marital status (*F* = 0.29, *p* = 0.835), and length of time in Taiwan (*r* = 0.05, *p* = 0.547) of the Indonesian caregivers were not significantly correlated with the overall workplace bullying scores, as shown in Table 1.

### 4.2. Overall and Sub-Scale Scores and Their Items for Workplace Bullying

A total of 109 (60.9%) Indonesian caregivers obtained more than 21 points from the workplace bullying score, indicating that they did not experience workplace bullying during the most recent six months (Table 1). The average workplace bullying score was 25.9 (±7.0) points, equivalent to a standard score of 24.7% as calculated the highest possible score of 105 points. This scale was divided into three sub-scales of workplace bullying. These were: intimidation toward a person (4 items), where the score was 5.4 (±2.0) points with 27.0% of the standard score; work-related bullying (7 items), where the score was 8.7 (±2.6) points with 24.9%; and personal-related bullying (10 items), where the score was 11.8 (±3.0) points with 23.6% in Taiwan (Table 2). 

### 4.3. Analysis of the Impact of Personal Characteristics, Sleep Quality, and Mental Health on Workplace Bullying among Indonesian Caregivers

A multiple regression analysis revealed that being a household caregiver (*β* = 0.14, *p* = 0.021), sleep quality (*β* = 0.18, *p* = 0.031), and mental health (β = 0.44, *p* < 0.001) were significantly correlated with the overall workplace bullying scores of the respondents. The explained variation values revealed 45%. The subjects who were household caregivers had a high sleep quality score (poor sleep quality) or had a high mental health score (poor mental health) and also tended to have a high overall workplace bullying score, as shown in Table 3.

## 5. Discussion

The results from this study indicated that the greater the age of the Indonesian caregiver, the more workplace bullying the individual was likely to experience. This finding was consistent with the research results pertaining to Vietnamese, Chinese, and Arabic-speaking foreign workers [31]. In this study, we found that Indonesian caregivers who worked at night were more often subject to workplace bullying, a result that was supported by the finding of other studies (i.e., bullying occurs with greater frequency when psychiatric nurses are working at night) [32]. Bullying tends to occur more often at night because the general shortage of night-time human power means that personnel must frequently work alone and because most people work during regular daytime hours or work different shifts, meaning that individuals may have less effective support systems at night [33].

In this study, 60.9% of Indonesian caregivers experienced bullying in a Taiwan workplace. This percentage was less than the Pakistani migrant workers working in the United Arab Emirates [9]. The standard score for workplace bullying of Indonesian caregivers of 24.7% in Taiwan was similar to that of foreign caregivers in Switzerland (25.6%) [34] and Indonesian employees in the workplace (25.0%) [15]. The most common form of workplace bullying among Indonesian caregivers in Taiwan was intimidation toward a person (standard score, 27.0%), followed by work-related bullying (24.9%). That of personal-related bullying (23.6%) was the lowest. However, the bullying of migrant workers from Myanmar in Thailand tended to take the form of personal-related bullying and intimidation toward a person by employers [8]. These differences may be attributable to differences in the work attributes of the foreign workers as well as the culture and customs of the host country [35].

In this study, we examined the relationships between the workplace bullying categories of foreign caregivers. Being a household caregiver, sleep quality, and mental health were closely related. For household caregivers who worked in the homes of their employers, there was typically no management system; in contrast, institutional caregivers are protected by Taiwan government assessment systems and have more effective management standards. This is supported by the previous finding that workplace bullying is less common at enterprises with better-developed standards and management systems [36]. Based on these results, the government should consider the actual situation of workplace bullying, sleep quality, and mental health among Indonesian household caregivers.

Studies have suggested that workplace bullying among employees tends to be accompanied by sleep problems and emotional distress [19]. The percentage of Indonesian caregivers with poor sleep quality (34.6%) in this study was lower than that of migrant workers from Myanmar in Malaysia (62.5%) [37]. In addition, in this study, 8.9% of the Indonesian caregivers surveyed had poor mental health; notably, there are rarely comparable data in the literature. In Taiwan, the percentages of the Indonesian caregivers with poor sleep quality and poor mental health were not high. The possible reasons could be the cultural background of speaking up, the pressure of giving socially desirable answers, or limitations of the convenience samples in general. However, these were the two most important factors influencing workplace bullying. Therefore, the empirical evidence of workplace bullying provided by this study can fill the knowledge gap of Indonesian caregivers in Taiwan.

Implications: The results of this study suggest that observation of sleep quality or mental health can be used to detect workplace bullying, identify potential problems, and improve and explore the quality of care and sleep quality or mental health of caregivers. Further, employers could be asked to emphasize the sleep status and mental health of foreign caregivers. The government could ask the employers to assess the sleep and mental status of the caregivers using brief psychological scales in community clinics. Physicians should have the responsibility to report to the government if they find foreign caregivers are not well. This suggestion not only respects those caregivers from overseas but also helps them to increase their quality of care for people living with disabilities. A large-scale study using a national or an international sample should be considered in terms of further research.

Limitations: First, the data from this study were collected at one hospital and a few organizations. The external validity of the findings may be limited to be applied to different settings. Second, the sample size was relatively small in this study. Last, there was a limit to the generalization of the results in a relatively biased sampling. This sample, therefore, may be limited in using the results to ascertain conclusions from this study.

## 6. Conclusions

This study investigated the relationships between the workplace bullying of foreign caregivers. Being a household caregiver, sleep quality, and mental health were closely related and were the most important factors influencing workplace bullying. To ensure that the frequency of workplace bullying decreases to the lowest possible level, there should be management and assessment systems for household caregivers and more attention should be paid to the sleep quality and mental health of Indonesian caregivers.

## Figures and Tables

**Table 1 ijerph-19-04909-t001:** Distribution and analysis of personal characteristics, sleep quality, and mental health with workplace bullying among Indonesian caregivers (*N* = 179).

Variables	*n*	M ± SD	Workplace	Bullying		
			M ± SD	*t/F/r*	*p*	Scheffé Test
Age		35.0 ± 6.8		*r =* 0.16	0.031	
22–30 years	46 (25.7%)		24.2 ± 5.9	*F =* 2.08	0.128	
31–40 years	90 (50.3%)		26.5 ± 7.0			
41–51 years	43 (24.0%)		26.7 ± 7.7			
Marriage				*F =* 0.29	0.835	
Married	103 (57.5%)		26.0 ± 6.9			
Unmarried	45 (25.1%)		25.5 ± 6.9			
Divorced	24 (13.4%)		26.0 ± 6.2			
Widowed	7 (3.9%)		28.1 ± 11.7			
Education level				*F =* 2.81	0.063	
Below middle school	74 (41.3%)		24.5 ± 6.1			
Junior high school or above	100 (55.9%)		27.0 ± 7.5			
Other or missing	5 (2.8%)		26.6 ± 5.8			
Length of time in Taiwan (months)		55.0 ± 32.0		*r =* 0.05	0.547	
<1 year (less than 12 months)	13 (7.3%)		24.6 ± 5.0	*F =* 0.45	0.815	
1–3 years (13–36 months)	43 (23.1%)		25.9 ± 7.7			
3–5 years (37–60 months)	51 (28.9%)		26.1 ± 6.6			
5–7 years (61–84 months)	38 (21.2%)		26.4 ± 7.6			
7–9 years (85–108 months)	25 (14.0%)		24.9 ± 6.7			
>9 years (109–156 months)	9 (5.0%)		28.3 ± 7.2			
Types of care services				*t =* −5.63	<0.001	
Institution	42 (23.5%)		22.5 ± 3.1			
Household	137 (76.5%)		27.0 ± 7.5			
Night-time work				*F =* 18.67	<0.001	(2) > (1)
No ^(1)^	91 (50.8%)		23.1 ± 3.9			
Yes ^(2)^	84 (46.9%)		29.0 ± 8.3			
On call ^(3)^	4 (2.2%)		26.0 ± 5.2			
Sleep quality		5.2 ± 3.9		*r =* 0.59	<0.001	
Poor (PSQI > = 6)	62 (34.6%)		30.6 ± 8.2	*t =* 6.25	<0.001	
Good (PSQI < 6)	117 (65.4%)		23.5 ± 4.7			
Mental health		2.5 ± 2.9		*r =* 0.66	<0.001	
Poor (BSRS-5 ≥ 6)	16 (8.9%)		28.6 ± 10.0	*t =* 5.24	<0.001	
Good (BSRS-5 < 6)	163 (91.1%)		24.8 ± 5.3			
Workplace bullying		25.9 ± 7.0				
Never (NAQ-R = 21)	70 (39.1%)					
Ever (NAQ-R > 21)	109 (60.9%)					

Note: (1) non night shift worker; (2) night shift worker; (3) had night shift work sometimes.

**Table 2 ijerph-19-04909-t002:** Overall and sub-scale scores and their items for workplace bullying (*N* = 179).

Variables	M ± SD	Standard Score (%)	Never	Sometimes	Monthly	Weekly	Daily
					*n* (%)		
Intimidation toward a person (4 items)	5.4 ± 2.0	27.0					
Repeated reminders of your errors or mistakes	1.6 ± 0.7	32.0	99 (55.3)	66 (36.8)	10 (5.5)	3 (1.7)	1 (0.6)
Excessive monitoring of your work	1.4 ± 0.8	28.0	129 (72.1)	40 (22.3)	3 (1.7)	3 (1.7)	4 (2.2)
Persistent criticism of your work and effort	1.3 ± 0.6	26.0	145 (81.0)	26 (14.5)	4 (2.2)	3 (1.7)	1 (0.6)
Being shouted at or being the target of spontaneous anger	1.2 ± 0.4	24.0	152 (84.9)	25 (13.9)	2 (1.1)	0 (0)	0 (0)
Work-related bullying (7 items)	8.7 ± 2.6	24.9					
Someone withholding information that affects your performance	1.5 ± 0.8	30.0	116 (64.8)	52 (29.1)	5 (64.8)	2 (1.1)	4 (2.2)
Being given tasks with unreasonable or impossible targets or deadlines	1.4 ± 0.6	28.0	125 (69.8)	47 (26.2)	4 (2.2)	3 (1.7)	0 (0)
Being ordered to do work below your level of competence	1.3 ± 0.7	26.0	148 (82.7)	17 (9.5)	11 (6.1)	2 (1.1)	1 (0.6)
Having key areas of responsibility removed or replaced with more trivial or unpleasant tasks	1.2 ± 0.4	24.0	149 (83.2)	28 (15.6)	1 (0.6)	1 (0.6)	0 (0)
Having your opinions and views ignored	1.2 ± 0.4	24.0	153 (85.4)	23 (12.8)	3 (1.7)	0 (0)	0 (0)
Being exposed to an unmanageable workload	1.2 ± 0.5	24.0	147 (82.1)	25 (13.9)	5 (2.8)	2 (1.1)	0 (0)
Pressure not to claim something that by right you are entitled to	1.1 ± 0.3	22.0	170 (94.9)	8 (4.4)	1 (0.6)	0 (0)	0 (0)
Personal-related bullying (10 items)	11.8 ± 3.0	23.6					
Being humiliated or ridiculed in connection with your work	1.4 ± 0.7	28.0	135 (75.4)	28 (15.6)	12 (6.7)	3 (1.7)	1 (0.6)
Being ignored or excluded	1.4 ± 0.6	26.0	123 (68.7)	52 (29.1)	3 (1.7)	0 (0)	1 (0.6)
Intimidating behavior such as finger-pointing, invasion of personal space, shoving, or blocking/barring the way	1.3 ± 0.7	26.0	136 (75.9)	38 (21.2)	0 (0)	2 (1.1)	3 (1.7)
Being ignored or facing a hostile reaction when you approach	1.2 ± 0.4	24.0	147 (82.1)	31 (17.3)	1 (0.6)	0 (0)	0 (0)
Hints or signals from others that you should quit your job	1.2 ± 0.5	24.0	151 (84.3)	22 (12.3)	5 (2.8)	1 (0.6)	0 (0)
Spreading of gossip and rumors about you	1.1 ± 0.4	22.0	159 (88.8)	19 (10.6)	0 (0)	0 (0)	1 (0.6)
Having insulting or offensive remarks made about your person (i.e., habits and background), your attitudes, or your private life	1.1 ± 0.4	22.0	160 (88.9)	17 (9.5)	1 (0.6)	0 (0)	1 (0.6)
Being the subject of excessive teasing and sarcasm	1.1 ± 0.4	22.0	165 (92.1)	13 (7.3)	1 (0.6)	0 (0)	0 (0)
Having allegations made against you	1.1 ± 0.3	22.0	171 (95.5)	7 (3.9)	0 (0)	1 (0.6)	0 (0)
Threats of violence or physical abuse or actual abuse	1.0 ± 0.2	20.0	174 (97.2)	5 (64.8)	0 (0)	0 (0)	0 (0)
Overall score (21 items)	25.9 ± 7.0	24.7					

Note: the standard score was equal to the score of overall workplace bullying or sub-scale divided by the highest possible score.

**Table 3 ijerph-19-04909-t003:** Analysis of the impact of personal characteristics, sleep quality, and mental health on workplace bullying among Indonesian caregivers (*N* = 179).

Variables	*B*	*SE*	*β*	*t*	*p*	95% C.I.	
						Lower	Upper
Personal characteristics							
Age	0.03	0.06	0.03	0.52	0.603	−0.09	0.16
Married ^1^	−0.10	0.85	−0.01	−0.12	0.904	−1.78	1.57
Junior high school or above ^2^	0.05	0.86	< 0.01	0.05	0.958	−1.66	1.75
Length of time in Taiwan	0.01	0.01	0.04	0.71	0.482	−0.02	0.03
Household caregivers ^3^	2.26	0.97	0.14	2.33	0.021	0.34	4.17
Night-time work ^4^	1.43	0.97	0.10	1.48	0.141	−0.48	3.34
Night-time on call ^5^	0.66	2.69	0.01	0.25	0.807	−4.65	5.97
Sleep quality	0.32	0.15	0.18	2.17	0.031	0.03	0.62
Mental health	1.05	0.20	0.44	5.33	<0.001	0.66	1.44
*Adjusted R²*			0.45				
*F*			16.96 (<0.001)				

Note: 95% C.I. indicates a 95% confidence interval. The multiple regression used personal characteristics, sleep quality, and mental health as independent variables to force entry into workplace bullying. Category-based personal characteristics first adopted a dummy variable, which included: ^1^ married (variable set to 1) and unmarried, divorced, or widowed (set to 0); ^2^ junior high school or above (set to 1) and below middle school and others (set to 0); ^3^ household caregivers (set to 1) and institutional caregivers (set to 0); ^4^ have night-time work (set to 1) and no night-time work (set to 0); and ^5^ night-time on call (set to 1) and no night-time work (set to 0).

## Data Availability

Not applicable.

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
