# Peer review of "Exploring the Workplace Bullying of Indonesian Caregivers and Its Influencing Factors in Taiwan"

_ijerph, 2022, doi:10.3390/ijerph19084909_

Round 1
Reviewer 1 Report
The topic is pertinent and current, understanding workplace bullying in any context is always a contribution to this area of knowledge and an added value for caregivers.
Abstract is well prepared and has all the parts so that the reader can understand the line of research, the methodology, its results and conclusions. However, authors are suggested to remove the numbering and that the text be in the form of paragraphs. The abstract score should be revised.
The introduction has an insufficient literature review to understand the variables under study, as well as suggest the authors who establish a relationship between the variables. The authors are suggested to formulate and present a research question to guide the line of investigation. At the end of the Introduction section it would be interesting for the authors to put a conceptual model and the relationship between the keywords. Considering that the authors followed a quantitative study, it would be interesting to formulate hypotheses to confirm the premises identified in the literature review.
The methodology followed is well explained and detailed for any investigator to be able to reproduce the investigation. However, it is suggested to the authors to characterize in a simpler way the data collection instrument (questionnaire), its origin and its validation in the study population. Authors are suggested to describe how the questionnaire was applied. It is suggested to the authors to better update the sample.
The results are well interpreted and the conclusions add knowledge to the area under study. In the conclusions, it is suggested to the authors to relate in more detail the different dimensions under study and that can be taken from the text of the questionnaire. Authors are suggested to identify the contributions of the study to the practical part, what teachings of the study we can introduce into the workplace. It is suggested to the authors to identify the contributions to the theoretical part. Authors are suggested to identify limitations and lines of investigation or future studies.
Author Response
Comments and Suggestions for Authors
The topic is pertinent and current, understanding workplace bullying in any context is always a contribution to this area of knowledge and an added value for caregivers.
Ans: Thank you.
Abstract is well prepared and has all the parts so that the reader can understand the line of research, the methodology, its results and conclusions. However, authors are suggested to remove the numbering and that the text be in the form of paragraphs. The abstract score should be revised.
Ans: We have removed the numbering from this article. Please see Page 1.
The introduction has an insufficient literature review to understand the variables under study, as well as suggest the authors who establish a relationship between the variables. The authors are suggested to formulate and present a research question to guide the line of investigation.
Ans: We have added information at the end of the Introduction. Please see Page 3.
At the end of the Introduction section it would be interesting for the authors to put a conceptual model and the relationship between the keywords.
Ans: The research team has discussed that this study has been finished. It is inappropriate to add extra information that we did not do before.
Considering that the authors followed a quantitative study, it would be interesting to formulate hypotheses to confirm the premises identified in the literature review.
Ans: At the end of the Introduction, the aim of this study is another format to state the possible hypotheses for this study. Most journal articles do not present hypotheses but instead use aim or research questions. Please see Page 3.
The methodology followed is well explained and detailed for any investigator to be able to reproduce the investigation.
Ans: Thank you.
However, it is suggested to the authors to characterize in a simpler way the data collection instrument (questionnaire), its origin and its validation in the study population. Authors are suggested to describe how the questionnaire was applied. It is suggested to the authors to better update the sample.
Ans: The reviewer has provided excellent comments here. We appreciate that. However, not all journals ask their authors to report in detail about your comments currently. The trend of scientific paper moves forward to new knowledge more, and basic information might not be needed.
The results are well interpreted and the conclusions add knowledge to the area under study.
Ans: Thank you.
In the conclusions, it is suggested to the authors to relate in more detail the different dimensions under study and that can be taken from the text of the questionnaire. Authors are suggested to identify the contributions of the study to the practical part, what teachings of the study we can introduce into the workplace. It is suggested to the authors to identify the contributions to the theoretical part. Authors are suggested to identify limitations and lines of investigation or future studies.
Ans: Thank you so much for your suggestions. We have fixed them based on your suggestions. Please see Page 3 and 8.
Reviewer 2 Report
This is a thesis emprically researching workplace bullying. It is necessary to consider the following points.
First, there is a limit to the generalization of the results in terms of bias sampling. It is necessary to suggest a way to overcome this weakness.
Second, if you are interested in the three types of bullying, comparisons between them should be the focus. For comparison between the three bullyings, it needs factor analysis which checks discriminant validity among the three factors.
Third, in the regression analysis that analyzes bullying, it is necessary to set the three kinds of bullying as dependent variables.
Fourth, in regression analysis, only individual demographic variables are currently suggested as variables affecting bullying. Based on the theoretical discussion, meaningful independent variables should be added.
Author Response
Comments and Suggestions for Authors
This is a thesis empirically researching workplace bullying. It is necessary to consider the following points.
First, there is a limit to the generalization of the results in terms of bias sampling. It is necessary to suggest a way to overcome this weakness.
Ans: The authors agree with the reviewer. We have added information in Page 8.
Second, if you are interested in the three types of bullying, comparisons between them should be the focus. For comparison between the three bullyings, it needs factor analysis which checks discriminant validity among the three factors.
Ans: Thank you for your advice. The study did not specifically compare the differences between three types of bullying. Thus, the factor analysis of the original scale was used as the basis for the concept of this study.
Third, in the regression analysis that analyzes bullying, it is necessary to set the three kinds of bullying as dependent variables.
Ans: Thank you for your advice. However, the study did not specifically compare the differences between the three types of bullying. Thus, the overall score for workplace bullying of regression analysis was used.
Fourth, in regression analysis, only individual demographic variables are currently suggested as variables affecting bullying. Based on the theoretical discussion, meaningful independent variables should be added.
Ans: Thank you for your suggestion. However, the independent variables are not only demographic variables but also sleep quality, and mental health variables. Please see Page 4 and Table 3.

Reviewer 3 Report
Dear authors,
Thank you very much for the opportunity to review your paper. Workplace bullying is a very relevant topic needing extensive research and speak-up. However in my opinion your paper needs some extensive revision in order to be properly understood by the international audience of this journal. Here are my suggestions for improvement. Good luck with your paper.
[2] Title: Please make clear that this paper is about Indonesian Caregivers in Taiwan, not in Indonesia
[13] Please remove numbers from the abstract. Please shorten background.
[19] Did you use Indonesian versions for all 3 questionnaires? In this case you should change to "versions"
[20] Why do you refer to those never bullied at all instead of those who are bullied? Please state briefly how many participants responded in your study.
[26] It reads to me that those who have sleeping disorders are at risk of being bullied, I think it´s just the other way round, those who are bullied (exposition) have an increased risk for sleeping disorders (symptom). Please clarify
[31] Why do you mention Japan in this extension? Is it relevant for your study. If not, please remove until line 43.
[49] You should start with this point followed by sentence starting in line 45.
[51] This is a relevant but maybe extreme example of another foreign country. Please add other available evidence of workplace bullying from other countries instead of section [31]-[43].
[97] repeated information about Taiwans legal situation from line 53, please remove
[112] Please state your a priori hypotheses made. In case this is purely explorative please clarify and adjust presentation of your results and your conclusions to this fact.
[115] This sentence is far too long and I do not understand what is meant
[125] This does not refer to subjects but to statistical analysis. You may start with sample size calculation and then mentioning what software was used. On what basis was sample size calculation performed? Did you aim to detect any significant differences by setting up those values. Please clarify and elaborate. How were subjects recruited (by email, mail, phone, in person?).
[130] did you use validated instruments or was Cronbach alpha detected during this survey for any of these questionnaires, please clarify.
[134] were collected
[159] using
[167] was your analysis adjusted for multiple testing, please clarify
[180]-[182] please remove
[185] females or women. As only women took place you have to make clear for the international audience: Are there any male caregivers from Indonesia at all? Can external validity of sample size be controlled by any statistics available from any census? You should consider speaking of female Indonesian caregivers in the title and abstract of your study.
[186] was married
[205] again, why do you stress out those who have not been bullied?
[221] Is it high sleep score or poor sleep quality, is it high mental health score or low mental health. If high sleep score means poor sleepp quality then please refer to clinical effects instead of misleading scores. Otherwise one could think that good sleep quality and bullying would correlate positively which does not seem to be true or rational.
[233] indicate that
[243] please rephrase this sections, first you mention 39,1% never been bullied then you state 24,7%.
[255] This conclusion cannot be made. Is poor sleep quality root for or effect of being bullied?
[269] You should discuss possible limitations of your studie more detailed. Could it be that sympotoms of bullying were not detected by your questionnaire as Indonesian caregives might react differently. Also please discuss cultural background of speaking up, pressure of giving socially desirable answers (esp. if survey took place at work place) and limitations of convencience samples in general.
[277] first sentence is not a conclusion but should be moved to results section.
[281] Again I would disagree that sleep quality influences risk of being bullied, but bullying influences risk of poor sleep quality.
Author Response
Comments and Suggestions for Authors
Thank you very much for the opportunity to review your paper. Workplace bullying is a very relevant topic needing extensive research and speak-up. However in my opinion your paper needs some extensive revision in order to be properly understood by the international audience of this journal. Here are my suggestions for improvement. Good luck with your paper.
Ans: Thank you.
[2] Title: Please make clear that this paper is about Indonesian Caregivers in Taiwan, not in Indonesia
Ans: Thanks for your suggestion. We have revised the tittle as “Exploring the workplace bullying of Indonesian caregivers in Taiwan and its influencing factors.” Please see Page 1.
[13] Please remove numbers from the abstract. Please shorten background.
Ans: Thanks for your advice. We have removed the number. Please see Page 1.
[19] Did you use Indonesian versions for all 3 questionnaires? In this case you should change to "versions"
Ans: Thanks for your suggestion. We did use Indonesian versions for all three questionnaires and have revised the term.
[20] Why do you refer to those never bullied at all instead of those who are bullied? Please state briefly how many participants responded in your study
Ans: Thanks for point this out. We have modified the number of those who are bullied. Please see Page 1, 5, and 7.
[26] It reads to me that those who have sleeping disorders are at risk of being bullied, I think it´s just the other way round, those who are bullied (exposition) have an increased risk for sleeping disorders (symptom). Please clarify
Ans: Thanks for your suggestion. We have revised the sentence. Please see Page 1.
[31] Why do you mention Japan in this extension? Is it relevant for your study? If not, please remove until line 43.
Ans: Thanks for your advice. We report other Asian countries in the Introduction because unemployment and increasing poverty have prompted many workers in developing countries. People who live in Northeast Asia, such as Taiwan or Japan, hire them to work with industries, including the healthcare industry.
[49] You should start with this point followed by sentence starting in line 45.
Ans: Thanks for your advice. We have fixed them based on your suggestions. Please see Page 2.
[51] This is a relevant but maybe extreme example of another foreign country. Please add other available evidence of workplace bullying from other countries instead of section [31]-[43]
Ans: Thank you so much for your suggestions. We did report other Asian countries in the Introduction because unemployment and increasing poverty have prompted many workers in developing countries. People who live in Northeast Asia hire them to work with industries, including the healthcare industry.
[97] repeated information about Taiwans legal situation from line 53, please remove
Ans: Thanks for your advice. We have removed the repeated information. Please see Page 3.
[112] Please state your a priori hypotheses made. In case this is purely explorative please clarify and adjust presentation of your results and your conclusions to this fact.
Ans: At the end of the Introduction, the aim of this study is another format to state the possible hypotheses for this study. Most journal articles do not present hypotheses but instead use aim or research questions. Please see Page 3.
[115] This sentence is far too long and I do not understand what is meant
Ans: Thank you so much for your suggestions. We have fixed them based on your suggestion. Please see Page 3.
[125] This does not refer to subjects but to statistical analysis. You may start with sample size calculation and then mentioning what software was used. On what basis was sample size calculation performed? Did you aim to detect any significant differences by setting up those values. Please clarify and elaborate. How were subjects recruited (by email, mail, phone, in person?
Ans: Thank you so much for your suggestions. We have fixed them based on your suggestions. Please see Page 3.
[130] did you use validated instruments or was Cronbach alpha detected during this survey for any of these questionnaires, please clarify.
Ans: Thank you. We have stated Cronbach alpha among each instrument. Please see Page 4.
[134] were collected
Ans: Thanks for your advice. Please see Page 3.
[159] using
Ans: Thanks for your advice. Please see Page 4.
[167] was your analysis adjusted for multiple testing, please clarify
Ans: We did analyze adjusted for multiple testing. Please see Page 4 and Table 3.
[180]-[182] please remove
Ans: Thanks for point this out. We have removed them. Please see Page 4.
[185] females or women. As only women took place you have to make clear for the international audience: Are there any male caregivers from Indonesia at all? Can external validity of sample size be controlled by any statistics available from any census? You should consider speaking of female Indonesian caregivers in the title and abstract of your study.
Ans: Thank you so much for your suggestions. We have modified the term. Please see Page 4.
[186] was married
Ans: Thanks for point this out. Please see Page 4.
[205] again, why do you stress out those who have not been bullied?
Ans: Thank you so much for your suggestions. We have fixed them based on your suggestions. Please see Page 5.
[221] Is it high sleep score or poor sleep quality, is it high mental health score or low mental health. If high sleep score means poor sleep quality then please refer to clinical effects instead of misleading scores. Otherwise one could think that good sleep quality and bullying would correlate positively which does not seem to be true or rational.
Ans: Thank you so much for your suggestions. We have fixed them based on your suggestions. Please see Page 6.
[233] indicate that
Ans: Thanks for point this out. Please see Page 7.
[243] please rephrase this sections, first you mention 39.1% never been bullied then you state 24.7%.
Ans: The 39.1% means those who never experienced bullying. We have revised the sentence. The second 24.7% means standard score for workplace bullying. Please see Page 7.
[255] This conclusion cannot be made. Is poor sleep quality root for or effect of being bullied?
Ans: Thank you so much for your suggestions. We have fixed them based on your suggestions. Please see Page 7.
[269] You should discuss possible limitations of your study more detailed. Could it be that symptoms of bullying were not detected by your questionnaire as Indonesian caregivers might react differently? Also please discuss cultural background of speaking up, pressure of giving socially desirable answers (esp. if survey took place at work place) and limitations of convenience samples in general.
Ans: Thank you so much for your suggestions. We have fixed them based on your suggestions. Please see Page 8.
[277] first sentence is not a conclusion but should be moved to results section.
Ans: Thank you so much for your suggestions. We have fixed them based on your suggestions. Please see Page 8.
[281] Again I would disagree that sleep quality influences risk of being bullied, but bullying influences risk of poor sleep quality.
Ans: Thank you so much for your suggestions. We have fixed them based on your suggestions. Please see Page 8.
Reviewer 4 Report
Comments
- The introduction does not state the contribution to the international literature.
- The data are limited to one hospital and few other organizations.
- The paper does not consider the potential heterogeneity in the estimated effects. Why? The relationships can differ significantly e.g., by age. The small sample size limits the conclusions.
- What is the external validity of findings that are presented in the paper?
- Are there practical policy conclusions for other settings?
Author Response
Comments and Suggestions for Authors
Comments
- The introduction does not state the contribution to the international literature.
Ans: We did report other Asian countries in the Introduction. Unemployment and increasing poverty have prompted many workers in developing countries, most from Southeast Asia, to seek work elsewhere. People who live in Northeast Asia, such as Taiwan or Japan, hire them to work with industries, including the healthcare industry.
- The data are limited to one hospital and few other organizations.
Ans: Thank you so much. We agree with you. We have added information that acknowledged it was a limitation in this study. Please see Page 8.
- The paper does not consider the potential heterogeneity in the estimated effects. Why? The relationships can differ significantly e.g., by age. The small sample size limits the conclusions.
Ans: Thank you so much. We agree with you. However, the foreign caregiver problem has been a severe issue in Taiwan, and it was hard to approach participants. Further, this was a clinical-driven study. We just wanted to explore the initial evidence for clinical practice.
- What is the external validity of findings that are presented in the paper?
Ans: Thank you so much for your excellent comment. The external validity of this study did exist. The reasons we have reported above in Page 3. We have added information that acknowledged it was a limitation in this study. Please see Page 8.
- Are there practical policy conclusions for other settings?
Ans: Thank you so much for your excellent comment. We have added information about practical policy in the Implication section. Please see Page 8.
Round 2
Reviewer 1 Report
The improvements made in the paper add a better reading and scientific understanding for the reader.
The improvements made guarantee the scientific standards of a paper.
Author Response
Dear Reviewer,
We have done our best to fix all the comments. Thank you for reviewing our work again.
Best regards,
The authors
Reviewer 2 Report
I did not find any reasons that change the first round decision
Author Response
Dear Reviewer,
Thank you for reviewing our work again. We have done our best to fix all the comments. We are sorry that you are not satisfied with our revision.
Best regards,
The authors
Reviewer 3 Report
Thank you very much for adressing my comments. Good luck.
Author Response

(The authors gave the same response as above.)

Reviewer 4 Report
I am happy with the revised paper.
Author Response

(The authors gave the same response as above.)
